# Role of the Fungus *Pneumocystis* in IL1β Pathway Activation and Airways Collagen Deposition in Elastase-Induced COPD Animals

**DOI:** 10.3390/ijms25063150

**Published:** 2024-03-09

**Authors:** Krishna Coronado, Carla Herrada, Diego A. Rojas

**Affiliations:** Instituto de Ciencias Biomédicas (ICB), Facultad de Ciencias de la Salud, Universidad Autónoma de Chile, Santiago 8910132, Chile; krishna.coronado@cloud.uautonoma.cl (K.C.); carla.herrada@cloud.uautonoma.cl (C.H.)

**Keywords:** *pneumocystis*, mucus, COPD, inflammation, IL1β, mucins, hypersecretion, fibrosis

## Abstract

Inflammation and mucus production are prevalent characteristics of chronic respiratory conditions, such as asthma and chronic chronic obstructive pulmonary disease (COPD). Biological co-factors, including bacteria, viruses, and fungi, may exacerbate these diseases by activating various pathways associated with airway diseases. An example is the fungus *Pneumocystis*, which is linked to severe COPD in human patients. Recent evidence has demonstrated that *Pneumocystis* significantly enhanced inflammation and mucus hypersecretion in a rat model of elastase-induced COPD. The present study specifically aims to investigate two additional aspects associated with the pathology induced by *Pneumocystis* infection: inflammation and collagen deposition around airways. To this end, the focus was to investigate the role of the IL-1β pro-inflammatory pathway during *Pneumocystis* infection in COPD rats. Several airway pathology-related features, such as inflammation, mucus hypersecretion, and fibrosis, were evaluated using histological and molecular techniques. COPD animals infected with *Pneumocystis* exhibited elevated inflammation levels, including a synergistic increase in IL-1β and Cox-2. Furthermore, protein levels of the IL-1β-dependent transcription factor cAMP response element-binding (CREB) showed a synergistic elevation of their phosphorylated version in the lungs of COPD animals infected with *Pneumocystis*, while mucus levels were notably higher in the airways of COPD-infected animals. Interestingly, a CREB responsive element (CRE) was identified in the Muc5b promoter. The presence of CREB in the Muc5b promoter was synergistically increased in COPD animals infected with *Pneumocystis* compared to other experimental groups. Finally, an increment of deposited collagen was identified surrounding the airways of COPD animals infected with *Pneumocystis* compared with the other experimental animal groups and correlated with the increase of *Tgfβ1* mRNA levels. These findings emphasize the role of *Pneumocystis* as a potential biological co-factor in chronic respiratory diseases like COPD or asthma, warranting new perspectives in the treatment of chronic respiratory diseases.

## 1. Introduction

Chronic obstructive pulmonary disease (COPD) is prevalent among adults and primarily associated with cigarette smoking. Chronic inflammation plays a pivotal role in the pathophysiology of the disease, contributing to several pathological features such as mucus hypersecretion, airway collapse, pulmonary emphysema, loss of club cells, goblet cell hyperplasia, and fibrosis [1,2]. Prolonged exposure of airways to pollutants triggers the release of pro-inflammatory cytokines, including TNFα, IL1β, and IL8 [3,4,5,6]. Notably, smokers have been reported to exhibit elevated levels of the cytokine IL1β compared to non-smokers [7]. Furthermore, an increase in IL1β levels has been detected in COPD patients through sputum and lung biopsy analyses [8,9]. Detection of IL1β levels in sputum has been suggested as an exacerbation predictor in COPD patients [10]. These elements associated with the IL1β pathway suggest a potential role in the pathophysiology of COPD influencing airway inflammation, mucus hypersecretion, and airway remodeling [11].

Moreover, another aspect observed in the pathophysiology of COPD is the emergence of fibrosis in small airways, a phenomenon linked to oxidative stress and the apoptosis of human fibroblasts [12,13]. Additionally, fibroblasts from small airways in COPD patients exhibit reduced proliferation, heightened expression of collagen, and increased levels of metalloproteinases MMP2 and MMP9 [14], along with an increment in the release of TGFβ1 [14,15]. Interestingly, the role of MM2 and MMP9 in the progression of COPD has been described, indicating an association with the degradation of extracellular matrix (ECM), increment of inflammation, and airway remodeling [16]. Furthermore, studies have reported that airway epithelial cells derived from COPD patients display an augmentation in epithelial-to-mesenchymal transition and elevated levels of TGFβ1 [15,17,18]. Nevertheless, other research has demonstrated that TGFβ1 is not the sole factor inducing fibrosis in small airways in COPD, as endothelin-1 (ET-1), which is a peptide secreted by the endothelial cells of airways, may also stimulate epithelial cells to activate fibroblasts, and it is markedly detected in the sputum of COPD patients. It is also believed that the release is due to the mechanical stress of airway closure [19,20].

The established exacerbation of COPD symptoms during viral and bacterial infections contrasts with the underexplored impact of pathogenic fungi on these exacerbations. *Pneumocystis* is an environmental fungus that infects specifically mammals, similar to *Pneumocystis jirovecii*, which infects humans, and *Pneumocystis carinii*, the fungus that infects rats [21]. *Pneumocystis* is the fungus causative of severe pneumonia in immunosuppressed individuals [22] and prevalent in infants [23]. In recent years, the prevalence of *Pneumocystis* has increased in non-HIV patients, who are associated principally with other immune diseases, solid organ transplantations, or autoimmune diseases. The complete guideline applied in the clinic to the treatment of *Pneumocystis-infected* patients may be revised in several articles [24]. It has been observed that the presence of *Pneumocystis* leads to an increment in inflammation and mucus hypersecretion in immunocompetent murine models. This induction leads to robust Th2 [25,26,27,28] or Th17 immune responses [29], sharing characteristics with COPD. Notably, *Pneumocystis* is highly prevalent in COPD patients and correlates with disease severity despite a low fungal burden [30,31,32]. Interestingly, this correlation persists in HIV models [33] and non-HIV patients [34], even when the *Pneumocystis* burden is low [30,31]. Moreover, limited animal models are exploring the interplay between *Pneumocystis* and COPD. Recently, the association between COPD and *Pneumocystis* infection in regard to disease severity has been demonstrated using an elastase-induced COPD rat model showing an increase in inflammation and mucus-related markers [35]. This new evidence supports the hypothesis that a mild *Pneumocystis* infection may amplify inflammation or other pathophysiological features, such as fibrosis in respiratory diseases like COPD, either directly or as a co-factor [36]. However, the mechanisms underlying this role remain elusive and it is necessary to explore other pathways, such as TGFβ1 or IL1β.

Hence, our objective was to investigate the contribution of *Pneumocystis* to the enhancement of inflammation, mucus secretion, and fibrosis related to the induction of the IL1β and TGFβ1 pathways within a COPD host. Additionally, we investigate the fibrotic response of the airways in this animal model. To achieve this, we conducted histological and molecular characterizations, assessing inflammatory, mucus, and fibrosis-related markers in an elastase-induced COPD animal model.

## 2. Results

### 2.1. Pneumocystis Status in an Elastase-Induced COPD Rat Model

The infection status of animals was assessed through two methods: cyst counting and qPCR burden determination. The animal groups evaluated in these experiments were one experimental group treated only with saline buffer and TMP-SMZ (trimethoprim-sulfamethoxazole) (TMS); one experimental group treated with elastase and TMP-SMZ (elastase; ELT); one experimental group treated with saline buffer and co-housing with *Pneumocystis*-infected animals (*Pneumocystis carinii*; Pc); and one experimental group treated with elastase and co-housing with *Pneumocystis*-infected animals (ELT-Pc). As shown in Figure 1A,B, the presence of *Pneumocystis* was evident solely in experimental animals co-housed with infected rats (Pc and ELT-Pc groups). These results were corroborated when cysts were quantified (Figure 1C). The burden of *Pneumocystis* was determined using qPCR assessments (Figure 1D), revealing the detection of the fungus exclusively in the infected experimental groups. No discernible differences were noted between the various infected animal groups (Pc and ELT-Pc groups).

### 2.2. The Increment of Inflammatory Cuffs in the Presence of Pneumocystis Is Not Potentiated in the Elastase-Induced COPD Rat Model

Chronic inflammation is a hallmark of COPD and manifests in both conducting airways and vessels. To measure inflammation, lung sections stained with hematoxylin and eosin (H/E) were examined under a 100× magnification, employing a semi-quantitative scoring approach, as outlined in the Section 4. The proportion of inflammatory cuffs surrounding bronchioles and vessels increased in the elastase-instilled (ELT and ELT-Pc) and infected (Pc) groups in comparison to the control group (Figure 2A,B). However, no discernible differences were noted between the ELT, Pc, and ELT-Pc experimental groups (Figure 2C,D). These results indicate that the presence of *Pneumocystis* does not potentiate the frequency of inflammatory cuffs. Nevertheless, the size of the cuffs or the composition of the infiltrates may be influenced during the infection.

### 2.3. Pneumocystis Increases Inflammatory Markers Associated with the IL1β Pathway in an Elastase-Induced COPD Rat Model

Recently, our laboratory group demonstrated an increment in the mRNA levels of several pro-inflammatory cytokines, such as *Il6*, *Il8*, and *Tnf*, in an elastase-induced COPD animal model infected with *Pneumocystis* [35]. However, it is essential to explore additional inflammatory pathways that may be activated under these conditions. To correlate histological findings with molecular mechanisms associated with other inflammatory pathways and identify differences between animal groups, the mRNA levels of Il1β were assessed through qPCR. Il1β mRNA levels were markedly higher compared to the other experimental groups, indicating a synergistic increase associated with the induction of COPD and *Pneumocystis* infection (Figure 3A). To evaluate the activity of pathways associated with IL1β, the mRNA of *Cox-2 (cyclooxygenase-2)* was measured, revealing a significant increase in the ELT-Pc animal group compared to the other animals (Figure 3B). Subsequently, the protein levels of the transcription factor associated with the IL1β pathway, cAMP response element binding (CREB) protein, were evaluated. The levels of phosphorylated CREB were elevated in the COPD group infected with *Pneumocystis* (Figure 3C,D and Appendix A), suggesting that the presence of the fungus *Pneumocystis* in the airways of COPD animals synergistically induced the activation of the IL1β pathway, leading to inflammation.

### 2.4. Pneumocystis Increases Mucus Secretion in the Airway Epithelium of Elastase-Induced COPD Rats by Induction of the IL1β Pathway

Mucus hypersecretion is recognized as a characteristic feature in various respiratory diseases, including COPD. In lungs infected with *Pneumocystis*, there is an additional increase in mucus within the airway epithelium due to the induction of several pathways, such as STAT6. However, many other pathways may be activated, such as IL1β. To detect mucus in these animals, sections stained with alcian-blue/periodic acid Schiff (AB/PAS) were examined under microscopy using a 100× magnification. Animals instilled with elastase and infected with the fungus (ELT and Pc) exhibited an increment in the presence of mucus in the airway epithelium compared to the control (Figure 4A,B). Notably, the presence of mucus was substantially higher in the ELT-Pc animal group compared to the ELT and Pc groups (Figure 4A,B). A significant increase in the mucus-stained area in the airway epithelium was noted in the ELT and Pc groups compared to control animals, while the ELT-Pc group displayed synergic increment compared to the ELT and Pc groups (Figure 4B).

Given the induction of the IL1β pathway in this animal model, and recognizing the association between this pathway and mucus secretion, we initially examined the presence of a cAMP response element (CRE) in promoters of Muc5ac and Muc5b. These mucin genes showed a clear increment in their expressions during the development of COPD induced by elastase and synergically incremented by the presence of *Pneumocystis* in the previous work [35]. A CRE sequence was only detected in the promoter of Muc5b (Figure 4C). Subsequently, we assessed the presence of phospho-CREB in the promoter of Muc5b of the experimental animals. As indicated in Figure 4D, a high presence of phospho-CREB was detected in the animal groups instilled with elastase and infected with *Pneumocystis* compared to the other groups. These findings suggest that *Pneumocystis* may induce the mucus secretion response in the airway epithelium of COPD animals through the IL1β pathway induction.

### 2.5. Pneumocystis Increases Collagen Deposition around the Airways in an Elastase-Induced COPD Rat Model

Previous studies have reported that *Pneumocystis* infection enhances collagen deposition around airways and vessels in the lungs of animals primarily infected with *Pneumocystis* [26]. Moreover, the activation of the IL1β pathway is associated with chronic inflammation and tissue fibrosis [37,38,39]. In this study, specific stains were utilized, as described in the Section 4, to measure collagen deposition. The level of collagen deposition around the airways of ELT and Pc animals exhibited a mild increase compared to the control group (Figure 5A,B). Interestingly, the airways of animals infected with the fungus and COPD displayed a synergistic increment compared to the ELT and Pc groups (Figure 5A,B). Additionally, COPD animals infected with *Pneumocystis* exhibited the highest increase in the mRNA levels of the pro-fibrosis gene *Tgfβ1* (Figure 5C). These findings suggest that the fibrosis induced in the airways of animals with COPD may be potentiated with the addition of a pathogen such as *Pneumocystis*, leading to an increase in the severity of the chronic disease.

## 3. Discussion

This work shows the role of a *Pneumocystis carinii* infection in elastase-induced COPD rats, and the presented results are a complementation of previous characterization of this animal model [35]. This model was chosen due to the simplicity of the application of elastase compared with cigarette smoke COPD induction. However, there are several limitations or disadvantages associated with the model in the development of COPD; for example, the administration of elastase leads to the development of emphysema lesions in the lungs of animals, which is only one aspect of COPD. However, through previous works, we have attempted to determine some relevant aspects of COPD, such as inflammation and mucus secretion, but the analysis of immune cells or specific cytokines requires a specific analysis associated with the specific animal model to generate the disease. For example, in cigarette smoke COPD-induced models, the altered cytokines may be different than in elastase models [40].

The main results presented previously in COPD animals infected with *Pneumocystis* documented a synergic increment of inflammation markers such as TNFα, IL6, IL8 and Cxcl2, an increment of mucus markers, such as Muc5ac and Muc5b, and Th2 transcription factors associated with mucus secretion, such as FoxA3 and Spdef. These results support the idea that this fungus may be a new element that triggers pathological processes in chronic respiratory diseases, leading to an increment of the severity of the symptoms. Moreover, the role of *Pneumocystis* has been proposed as a key factor in the first steps of the development of COPD using immunosuppressed animals [41]. In addition, studies of human COPD patients infected with *Pneumocystis jirovecii* (the causative agent of human *Pneumocystis* infection) showed that the detection of this fungus correlates with higher pro-inflammatory cytokines levels, such as TNFα, IL6, and IL8 [32]. Additionally, *Pneumocystis jirovecii* colonization has been detected in COPD patients correlating with acute exacerbations [42] and has been related to the induction of a strong Th1 inflammatory response in COPD patients [43] or Th2 and Th17 inflammatory responses [26,28,29,44]. However, in examining all presented data, it is plausible that a mixed response could be observed in the progression of the disease in the COPD animals infected with the fungus *Pneumocystis*, explaining the increased severity of the disease, especially those related to the synergic inflammatory and mucus secretion response. Therefore, exploration of other pathways must be developed to understand the role of *Pneumocystis* as a co-factor in the increment of the severity of COPD, and thus, in this work, we documented the role of other pathways, such as IL1β and TGFβ1, which have been proposed as key factors and have a potential contribution to the increment of the severity of the disease in these infected animals. The alteration of these pathways in the context of *Pneumocystis* infection in COPD animals may explain the increment of the pathologic features observed in the results of the work, such as high inflammation, high mucus secretion, or collagen deposition.

In this work, we showed that the presence of inflammation cuffs in the lungs showed results comparable to the previously described [35]. Interestingly, the number of cuffs between animal groups infected (Pc) and COPD infected (ELT-Pc) did not show a significant difference, indicating that not only must the number of cuffs be evaluated, but the size and the cell components of each cuff must to be coherent with the analysis of cytokines previously presented [35]. Another result of this work is the increment of IL1β mRNA levels indicating the activation of the pathway. Levels of IL1β were synergically increased in COPD animals infected with *Pneumocystis*. In addition, one of the most important activators of the pathway, Cox-2, showed a substantial increment in its mRNA levels in COPD animals infected with the fungus. These results are consistent with the role of the IL1β pathways during the development of COPD in patients where the activation of the pathway leads to an increment of IL1β and correlates with the severity of the disease and exacerbations [45,46]. Moreover, it is important to mention that IL1β plays an important role in increasing inflammation in the airways of COPD patients and animals, leading to the development of inflammation and emphysema in the lungs [9,47,48,49,50]. The increment of IL1β levels is consistent with the finding that, in other *Pneumocystis*-infected animal models, other pro-inflammatory cytokines are increased, correlating with the presence of the fungus in the lungs, such as Tnfα, Cxcl2, IL6, or IL8 [26,28,35]. Additionally, the increment of mRNA of Cox-2 is consistent with their role in COPD described previously, where the levels of Cox-2 are increased in COPD [51] and the inhibition of Cox-2 function using celecoxib reduces the inflammation in rats with emphysema induced with cigarette smoke [52].

Another result of this work is the increment of mucus in the airways of COPD animals. Interestingly, a higher increment was observed in COPD animals infected with *Pneumocystis*. This result is consistent with the previous characterization of this animal model, as well as with previous studies where the association between *Pneumocystis* infection and mucus hypersecretion was stated, including the synergic increment of Muc5ac and Muc5b levels by the presence of *Pneumocystis* [28,35,53]. There are several examples of evidence showing the relation between the IL1β pathway and mucus secretion, where the increment of IL1β is associated directly with high levels of expression of Muc5ac and Muc5b in airways [54,55,56,57]. Interestingly, in this work we documented that the expression of one of the most important mucins of the lung, Muc5b, could be regulated with the IL1β pathway through the transcription factor CREB, indicating a strong association of this transcription factor with Muc5b promoter in COPD animals infected with *Pneumocystis* compared with the other experimental groups. These results were complemented by the mild increment of protein levels of the phosphorylated version of CREB in the animals with COPD and infected with the fungus. These results are consistent with the role of CREB recently documented in club cells, which is related to the increased mucus secretion [58] but associated with the regulation of other transcription factors, such as FoxA2 and Spdef, which have been associated with the regulation of mucin expression [28,59] and not directly with the mucin promoters. Other studies showed that Muc5ac expression is upregulated by IL1β via NF-κB and HIF-1α pathways [60]. Further studies must be performed to evaluate the role of CREB in modulating mucin expression and how the fungus *Pneumocystis* may trigger this regulation.

Additionally, in this work, we also have documented the increase in collagen deposition around airways having a higher increment in COPD animals infected with *Pneumocystis*. Collagen accumulation has been described previously in *Pneumocystis* primary infection animal models correlating the increment of collagen deposition with the presence of *Pneumocystis* in those animals [26]. Interestingly, the IL1β pathway has been associated with the development of lung fibrosis, stimulating collagen synthesis and the proliferation of fibroblasts [61,62] and promoting the expression of the pro-fibrosis marker TGFβ1 [63,64]. In vivo studies have demonstrated the role of IL1β in leading to a fibrotic response in the lungs [37,39], while other studies have indicated that the neutralization of IL1β may result in a deficient fibrotic response in mice [65], underscoring the active role of this cytokine in the induction of fibrosis. Additionally, the accumulation of collagen around the airways in the progression of COPD has been related to abnormal mechanics of the airways [66]. In addition, the changes in the features of the collagen fibers during emphysema development have been associated with the impairment of lung function that is associated with the clinical manifestations linked with COPD, such as airway obstruction, dyspnea, and coughing. Further experiments must be performed to elucidate the effect of the increment of collagen deposition in the airways of COPD animals infected with *Pneumocystis* and how this fibrotic feature could be related to the putative worsening of pulmonary function parameters having several clinical implications in patients.

Finally, the high prevalence of *Pneumocystis* in the community and in COPD patients, coupled with the high evidence from animal models [35,67], suggest that this fungus could potentially exacerbate the respiratory symptoms and collaborate with other insults that trigger these chronic respiratory diseases, such as COPD and asthma. For example, in studies evaluating the composition of the mycobiota in asthma patients, *Pneumocystis* was identified as an important component and associated with children with severe asthma [68]. These findings could lead to new studies that report the disturbing effects of *Pneumocystis* on normal airway mycobiota and its implications in regard to the severity of other chronic respiratory diseases. Additionally, it is possible to identify the role of other pathways implied in lung function, such as the retinoic acid pathway, which is related to tissue repair, and deficiency of vitamin A potentially leading to the development of emphysema [69]. Interestingly, molecules derived from retinoic acid have been proposed, such as anti-fungal drugs, including promising effects against *Pneumocystis* [70].

Other evaluations could be performed to identify the relationship between the presence of this fungus and the symptomatology in patients to determine if this relation could be associated with the worsening of chronic respiratory disease features, such as chronic inflammation and mucus secretion. Therefore, it is crucial to continue these investigations to elucidate the clinical implications of the involvement of *Pneumocystis* in the development of chronic respiratory disease.

## 4. Materials and Methods

### 4.1. Ethics

Testing with animals was conducted in facilities within the Faculty of Medicine of the University of Chile under the protocol CBA#1110 approved by the Institutional Committee for Care and Use of Animals (CICUA) (certification N° 19336-MED-UCH). Experiments were performed according to the Animal Protection Law 20,380 of Chile and the Guide for the Care and Use of Laboratory Animals (8th Edition, National Academies Press, Washington, DC, USA).

### 4.2. Elastase-Induced COPD Animal Model and Pneumocystis Infection

The animal model performed in this work is substantially the same as those previously published for our group, with few modifications. Briefly, Sprague Dawley female rats of 300 g weight derived from a single colony were arranged in groups of 3 rats per cage. Then, 1 U of elastase (SIGMA, St. Louis, MO, USA) per animal was instilled under anesthesia conditions in half of the animals. The other animals were instilled with saline buffer. Tylosine (1 g/L) was added to the drinking water for 4 weeks after instillation to prevent bacterial infections. Delivery of *Pneumocystis* was conducted 4 weeks after instillation by co-housing using *Pneumocystis carinii*-infected animals [26]. These animals were put in each cage for 1 week in a co-housing ratio of 1:3. Prevention of *Pneumocystis* infection was performed in half of the experimental groups by the addition of trimethoprim-sulfamethoxazole (TMP-SMZ; 80 mg TMP and 400 mg SMZ per 5 mL) 15 mL/L to the drinking water. In summary, one experimental group was treated only with saline buffer and TMP-SMZ (TMS); one experimental group was treated with elastase and TMP-SMZ (ELT); one experimental group was treated with saline buffer and co-housing with *Pneumocystis*-infected animals (Pc); and one experimental group was treated with elastase and co-housing with *Pneumocystis*-infected animals (ELT-Pc). A summary of the animal model is presented in Figure 1A. The number of animals per group was n = 5. Animals were euthanized under deep anesthesia 14 weeks after instillation. Lungs were extracted from the thorax and washed with sterile PBS. The left lobule was separated and fixed in 3.7% formalin buffered with PBS (pH 7.4). Right lobules were quickly frozen at −80 °C.

### 4.3. Pneumocystis Rapid Purification and Cysts Quantification

A total of 200 mg of frozen lungs were finely chopped into several pieces and combined with 10 mL of sterile, cold PBS. The mixtures were stirred using a magnetic stirrer for 1 h at 4 °C. Subsequently, the liquid underwent filtration through sterile gauze to eliminate tissue debris. Then, the liquid was passed three times through a 3 mL 21-gauge syringe. The resulting material was collected through cold centrifugation for 10 min at 500× *g*. The collected material was washed with 1 mL of PBS and transferred to a new 1.5 mL microcentrifuge tube. Afterward, the tubes underwent cold centrifugation for 10 min at 500× *g*, and the collected material was resuspended in 400 μL of cold PBS. For analysis, 1 μL of each purification was diluted with 50 μL of PBS, and 10 μL of this dilution were examined to count the number of cysts by cresyl violet staining. In brief, purified Pneumocystis cells were placed on a clean slide and fixed at a high temperature (70 °C for 5 min). A drop of 1% cresyl violet (ab246817, Abcam, Cambridge, UK) was then applied to the fixed cells, and, after 5 min, slides were carefully washed three times with distilled water. Subsequently, the slides were dried in an oven at 70 °C until no water was detected. Following this, slides were mounted and assessed under optical microscopy. The number of cysts was quantified using 20 fields per animal at a total magnification of 1000×.

### 4.4. Detection of Pneumocystis Burden

Quantitative PCR was employed to detect *Pneumocystis carinii* burden, using total genomic DNA extracted from 20 mg of freshly frozen lung tissue with the Genomic DNA Isolation Kit (Norgen Biotek, Thorold, ON, Canada). An amount of 200 ng of genomic DNA served as a template for amplifying a segment of the fungus dhfr gene under standard PCR conditions, involving an initial denaturation step of 15 min at 95 °C and 40 cycles of 15 s at 95 °C, 15 s at 55 °C, and 15 s at 72 °C. The PCR products were compared with the identical product inserted in a pGEM-T Easy plasmid (Promega, Madison, WI, USA). All experiments were conducted using Brilliant II SYBR Green QPCR Master Mix (Agilent, Santa Clara, CA, USA) and the AriaMx qPCR instrument (Agilent, Santa Clara, CA, USA). The primer sequences employed in this detection are detailed in Table 1.

### 4.5. Histology

The study involved the utilization of 5 μm thick paraffin-embedded sections from formalin-fixed lungs that had been immersed in formalin for 48 h. These sections were subsequently deparaffinized and stained using either hematoxylin and eosin (H&E) or Alcian blue/periodic acid Schiff (AB/PAS) staining techniques, following established procedures. To assess inflammation, we determined the proportion of peribronchiolar and perivascular cuffs by calculating the percentage of inflamed airways or vessels observed within each lung section from individual rats (n = 5). The AB/PAS-stained sections were employed to evaluate the extent of bronchiolar epithelium occupied by mucus, with all procedures executed according to the manufacturer’s guidelines (DIAPATH, Martinengo, Italy). The measurements were conducted using Image J software version 1.53 (MCBI, Bethesda, MD, USA). The percentage of the stained area was determined by calculating the ratio of the AB/PAS-stained area to the total epithelium area, defined as the region between the luminal surface and the basal membrane. Collagen deposition was determined using Masson Trichrome staining (DIAPATH, Martinengo, Italy) following the manufacturer’s protocol. The collagen deposition was determined as the ratio between the stained area surrounding the airway epithelium and the perimeter of the airway. Fourteen bronchioles were analyzed in all the staining experiments. All images were captured using an Olympus BX60 microscope and Image Pro Plus version 5.1.0 software (Media Cybernetics Inc., Rockville, MD, USA).

### 4.6. Gene Expression Determinations

To assess changes in mRNA levels related to inflammation, mucus, and fibrosis, we initiated the process by extracting total RNA from fresh-frozen lung tissue. We used 20 mg of lung tissue, which was homogenized in the presence of 500 μL of RNA solv reactive (Omega Biotek, Norcross, GA, USA). Then, the reactions were subjected to vortex mixing in the presence of 100 μL of chloroform. The resulting organic and aqueous phases were separated through cold centrifugation for 15 min at 14,000× *g*. Subsequently, the aqueous phase was transferred to a clean microcentrifuge tube and combined with 250 μL of isopropylic alcohol. These reactions were incubated at −20 °C for an hour. RNA was collected via another round of cold centrifugation for 15 min at 14,000× *g*. The RNA pellets were washed by the addition of 70% ethanol. The RNAs were then suspended in 50 μL of nuclease-free water, quantified, and stored at −80 °C. For each RNA, 2 μg were incubated with 1 unit of DNase (Thermo Fisher Scientific, Waltham, MA, USA) and their specific buffer in a final volume of 10 μL for 30 min at 37 °C. To inactivate the enzyme, 2 μL of 25 mM EDTA was added to the mixture and incubated at 65 °C for 10 min after the DNAse treatment. Following this, each RNA was incubated for 5 min at 70 °C with 0.5 μg of random hexamer primers (Promega, Madison, WI, USA) in a final volume of 12.5 μL. The reactions were then chilled on ice for 5 min, and nucleotides, buffer, RNAse inhibitor, and reverse transcriptase were added according to the M-MLV protocol (Promega, Madison, WI, USA) to reach a final volume of 25 μL. These mixes were incubated for 90 min at 37 °C, with an additional step of 10 min at 70 °C to inactivate the enzyme. The resulting single-stranded cDNAs were diluted at a 1:4 ratio, and 2 μL were used as templates in qPCR experiments. The primers utilized in this study are listed in Table 1. All experiments were carried out using a Brilliant II SYBR Green QPCR Master Mix (Agilent, Santa Clara, CA, USA) and the AriaMx qPCR instrument (Agilent, Santa Clara, CA, USA). The PCR reactions followed these conditions: an initial hot start at 95 °C for 12 min, followed by 45 cycles of 20 s at 95 °C, 20 s at 58 °C, and 20 s at 72 °C. Actin served as the internal control for all gene markers. Results were expressed as fold changes, normalized by actin, and referenced to the control group using the 2^−ΔΔCt^ method [71].

### 4.7. Protein Extractions

Proteins were extracted from 40 mg of fresh-frozen lung tissues. These tissues underwent homogenization in 500 μL of modified RIPA buffer (comprising 50 mM Tris at pH 7.4, 1% NP-40, 0.5% sodium deoxycholate, 150 mM NaCl, and 1 mM EDTA), supplemented with a total protease inhibitor cocktail (Roche, Mannheim, Germany). Subsequently, an additional 300 μL of RIPA buffer and sodium dodecyl sulfate at a final concentration of 0.01% were introduced to the extractions. These mixtures were maintained on ice for one hour. Then, the extractions were passed through a 21-gauge syringe three times. The extractions underwent cold centrifugation twice at 14,000× *g* for 15 min and supernatants were stored. Protein quantification was carried out using the Bradford Protein Kit (BioRad, Hercules, CA, USA). Samples from each lung tissue were prepared by mixing them with Laemmli buffer (BioRad, Hercules, CA, USA) supplemented with 10% 2-mercaptoethanol and heated at 95 °C for 10 min. Then, proteins were stored at −80 °C.

### 4.8. Western Blotting Determinations

An amount of 30 μg of each protein sample extract were assessed to determine changes in the levels of CREB, phospho-CREB, and actin. The proteins were separated through electrophoresis in a 12% polyacrylamide gel, 1 mm thick, supplemented with 0.1% sodium dodecyl sulfate. The samples were electrophoresed until the running front reached 4 cm from the wells. Subsequently, the proteins were transferred to a PVDF membrane (Amersham Biosciences, Amersham, UK) using a Mini Trans-Blot Electrophoretic Transfer Cell (BioRad, Hercules, CA, USA) for one and a half hours at 350 mA. Following the transfer, the membranes were blocked with 5% non-fat milk in TBS for two hours at room temperature. After washing the membranes, they were probed with anti-CREB (ab32096, Abcam, Cambridge, UK), anti-phospho-CREB (ab32515, Abcam, Cambridge, UK), or anti-actin (sc-8432, Santa Cruz Biotechnology, Dallas, TX, USA) overnight at 4 °C at dilutions of 1:2000 in 1% non-fat milk prepared in TBS. The primary antibodies were detected using a secondary antibody conjugated with HRP enzyme activity (sc-516102, Santa Cruz Biotechnology, Dallas, TX, USA, or ab6721, Abcam, Cambridge, UK), and the complexes were visualized through chemiluminescence using the SuperSignal West PICO Plus kit (Thermo Fisher Scientific, Waltham, MA, USA). The protein bands were quantified using Image J software version 1.53 (NCBI, Bethesda, MD, USA). The results were expressed relative to the internal control, actin protein.

### 4.9. Chromatin Immunoprecipitation (ChIP)

Fresh lung samples (30 mg aliquots) underwent homogenization in 300 mL of cold PBS 1X with a final concentration of 1% formaldehyde. Following this, the samples were incubated for 10 min at room temperature. The crosslinking reactions were stopped by adding glycine to a final concentration of 125 mM and incubating for an additional 5 min. The fixed material was then collected through centrifugation at 2000× *g* for 5 min at 4 °C. Pellets were washed by adding 500 µL of cold PBS 1X. The collected pellets were suspended in 500 µL of ChIP lysis buffer (10 mM Tris pH 8.0, 10 mM NaCl, 3 mM MgCl_2_, 0.5% NP-40, complete protease inhibitor cocktail), incubated on ice for 5 min and centrifuged at 4 °C for 5 min at 2000× *g*. The collected material was then suspended in 300 µL of ChIP nuclear lysis buffer (50 mM Tris pH 8.1, 1% SDS, 5 mM EDTA, complete protease inhibitor cocktail). Nuclei lysis and chromatin shearing were carried out through sonication, and the sheared chromatin was separated from nuclear debris by centrifugation at 4 °C for 5 min at 2000× *g*. For each sample, 50 µL of the supernatant (chromatin) was pre-cleared with 30 µL Protein A/G-agarose beads (sc-2003, Santa Cruz Biotechnology, Dallas, TX, USA), complete protease inhibitor cocktail, and ChIP dilution buffer (16.7 mM Tris pH 8.1, 0.01% SDS, 1.1% Triton X-100, 167 mM NaCl). The beads were separated from the chromatins by centrifugation at 5000× *g* for 5 min at 4 °C. Subsequently, 2 µg of anti-P-CREB (ab32515, Abcam, Cambridge, UK) or IgG (sc-2027, Santa Cruz Biotechnology, Dallas, TX, USA) were added to the chromatins, and the mixes were incubated overnight. Then, 30 µL of blocked Protein A/G-agarose beads were added to each sample and incubated at 4 °C for 4 h. The beads were collected by centrifugation for 5 min at 5000× *g* at 4 °C, and the supernatants were discarded. The beads were washed once with 1 mL of low salt buffer, high salt buffer, and LiCl_2_ buffer, and then twice with TE buffer. Each wash involved a 5 min incubation at 4 °C, followed by centrifugation at 5000× *g* for 5 min at 4 °C. After all washes, the beads were incubated at room temperature with 125 µL of elution buffer (1% SDS, 0.1% NaHCO_3_) for 15 min. Supernatants were collected by centrifugation, discarding the beads. Subsequently, 6.3 μL of 4 M NaCl were added to each sample until a final concentration of 0.192 M was reached. Des-crosslinking was conducted at 65 °C overnight. Protein digestion was carried out at 50 °C for 2 h using Proteinase K (Thermo Scientific, Waltham, MA, USA). The precipitated DNA was collected using the FavorPrep GEL/PCR purification kit (Favorgen, Vienna, Austria), and the DNA was suspended in a final volume of 50 µL. Aliquots of 3 µL were analyzed by qPCR to amplify the CREB binding site in the rat Muc5b promoter, with primer details provided in Table 1.

### 4.10. Statistics

We conducted all analyses using GraphPad Prism 10 (GraphPad Software Inc., San Diego, CA, USA). Results were presented as the mean ± standard deviation (SD). Each group consisted of 5 animals. The data distribution was assessed using the Shapiro–Wilk test, and distinctions between experimental animal groups were evaluated through one-way ANOVA followed by Tukey’s multiple comparison test. A statistically significant result was defined as a *p*-value less than 0.05.

## Figures and Tables

**Figure 1 ijms-25-03150-f001:**
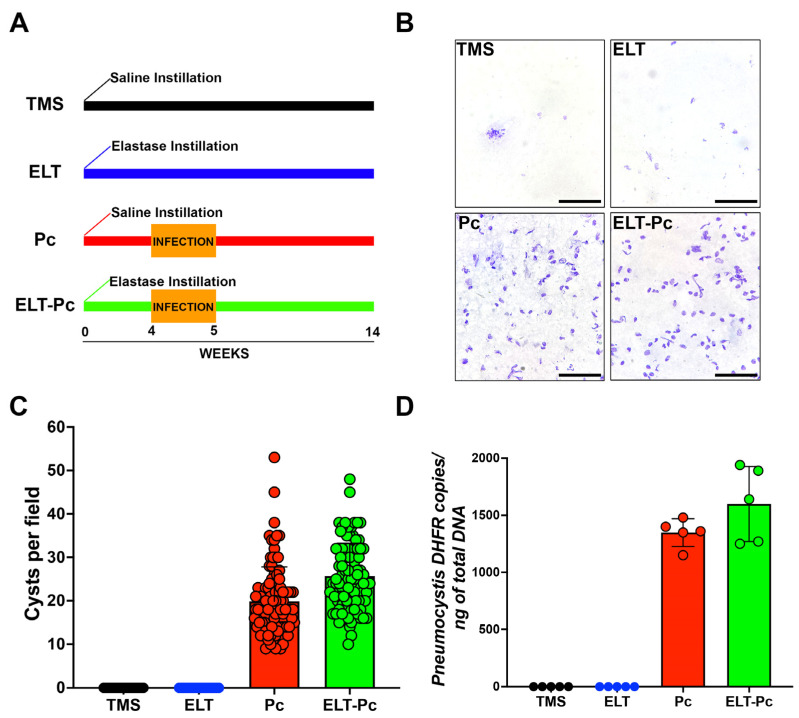
*Pneumocystis* status in an elastase-induced COPD rat model. (**A**) Diagram of the animal model outlining the four experimental groups: TMS, Elastase (ELT), *Pneumocystis* infection (Pc), and Elastase-*Pneumocystis* infection (ELT-Pc). The experiment spanned 14 weeks, encompassing one week of cohabitation with *Pneumocystis*-infected rats, occurring four weeks after elastase instillation in the *Pneumocystis* groups. (**B**) Representative images displaying purified *Pneumocystis* from the lungs of rats in all experimental groups stained with cresyl violet. The lungs of TMS and ELT groups underwent the same treatment as those from infected animals. The scale bar represents 20 µm, and microscopic analyses were conducted at 1000× magnification. (**C**) The number of *Pneumocystis* cysts was quantified using 100 fields per animal group. (**D**) The *Pneumocystis* burden was assessed through qPCR, quantifying the copies of the *dhfr* gene relative to the total DNA in lung samples collected from animals in each experimental group.

**Figure 2 ijms-25-03150-f002:**
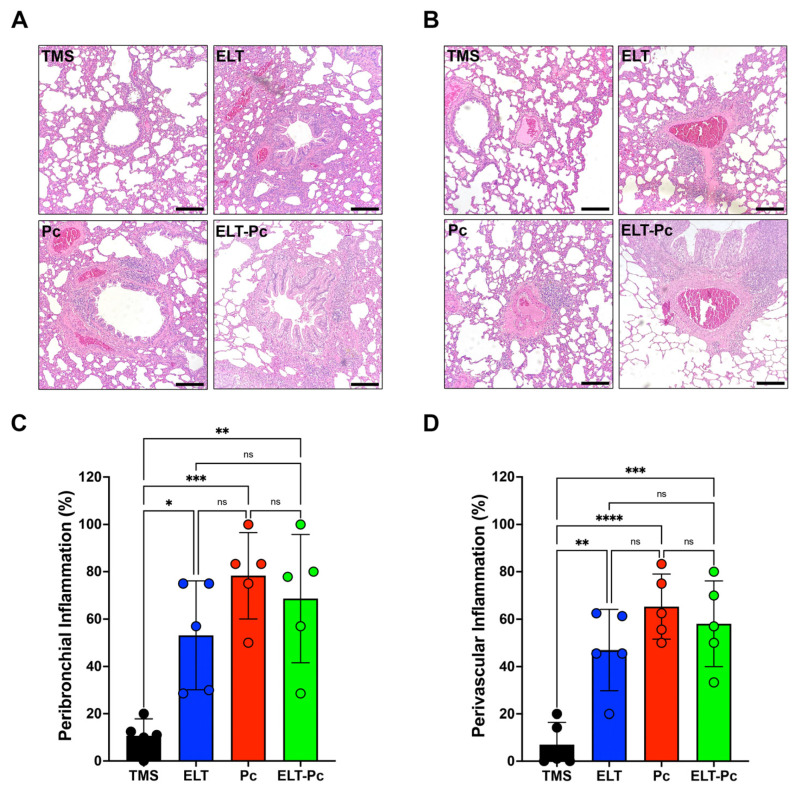
Quantification of peribronchiolar and perivascular inflammation cuffs in the elastase-induced COPD model infected with *Pneumocystis*. Hematoxylin-eosin-stained lung sections from the experimental groups of rats were examined microscopically at 100× magnification to identify the presence of inflammation cuffs. Peribronchiolar (**A**) and perivascular (**B**) inflammation cuffs were indicated in the images. The bar indicates 100 μm. The quantification of detected cuffs in the images utilized a scoring system detailed in the Section 4. The quantification of peribronchiolar (**C**) and perivascular (**D**) inflammatory cuffs is presented. Data are expressed as mean ± SD and analyzed by one-way ANOVA. Ns = non significant; * = *p* < 0.05; ** = *p* < 0.01; *** = *p* < 0.001; **** = *p* < 0.0001.

**Figure 3 ijms-25-03150-f003:**
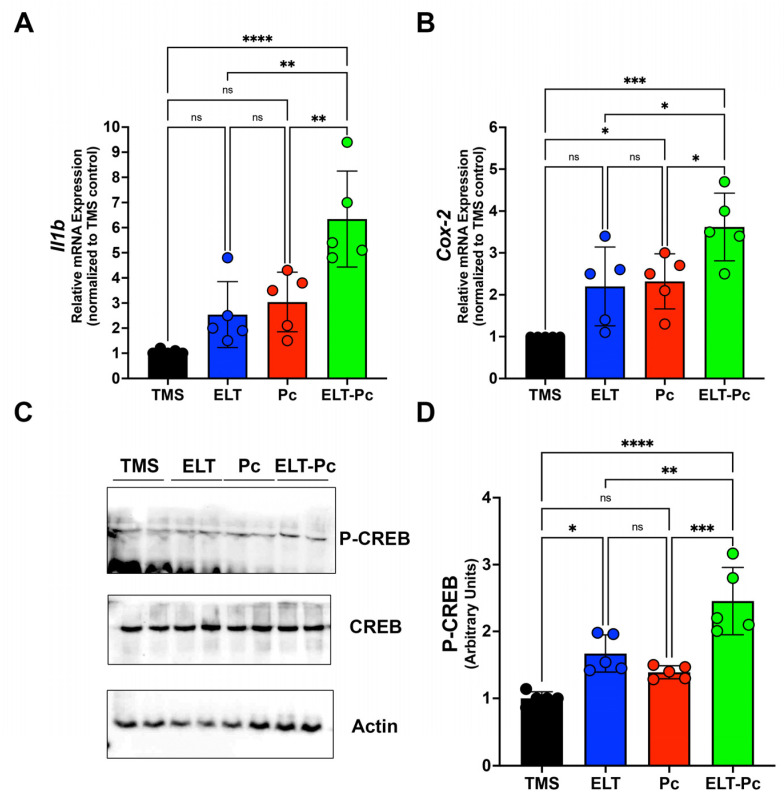
Characterization of IL1β cytokine pathway in the elastase-induced COPD animal model infected with *Pneumocystis*. (**A**) *Il1β* cytokine and (**B**) *Cox-2* mRNA levels were evaluated by qPCR. Data were determined using the 2^−DDCt^ method; actin levels were used as internal control and normalized to the TMS control animal group. Data are expressed as mean ± SD and analyzed by ANOVA. Ns = non-significant; * = *p* < 0.05; ** = *p* < 0.01; *** = *p* < 0.001; **** = *p* < 0.0001. (**C**) Protein levels of transcription factor CREB and the phosphorylated version were determined. Actin was used as an internal control. (**D**) Quantification of Western blots was determined by normalizing data with CREB and actin protein levels and protein from the TMS animal group. Data are expressed as mean ± SD and analyzed by ANOVA. Ns = non-significant; * = *p* < 0.05; ** = *p* < 0.01; *** = *p* < 0.001; **** = *p* < 0.0001.

**Figure 4 ijms-25-03150-f004:**
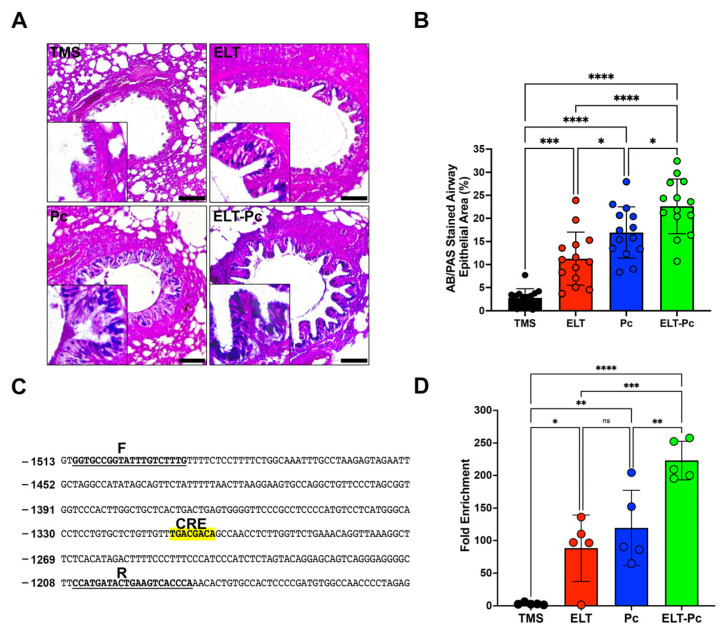
Relation between mucus secretion and IL1β pathway activation in the elastase-induced COPD animal model infected with *Pneumocystis*. (**A**) Representative images of lung sections from the experimental animal groups stained with alcian-blue/periodic acid Schiff (AB/PAS). Microscopic magnification was set at 100×. Insets indicate the detail of stained airway epithelial cells. (**B**) The stained area of the airway epithelium was quantified following the procedure outlined in the Section 4. (**C**) Muc5b promoter section highlighting the putative CREB response element (CRE). “F” and “R” denote the sequences of the primers employed in the analysis of the binding of phosphorylated CREB transcription factor. (**D**) Quantification of the chromatin immunoprecipitation (ChIP) evaluating the binding of phosphor-CREB to the Muc5b promoter. Data are expressed as mean ± SD and analyzed by ANOVA. Ns = no-significative; * = *p* < 0.05; ** = *p* < 0.01; *** = *p* < 0.001; **** = *p* < 0.0001.

**Figure 5 ijms-25-03150-f005:**
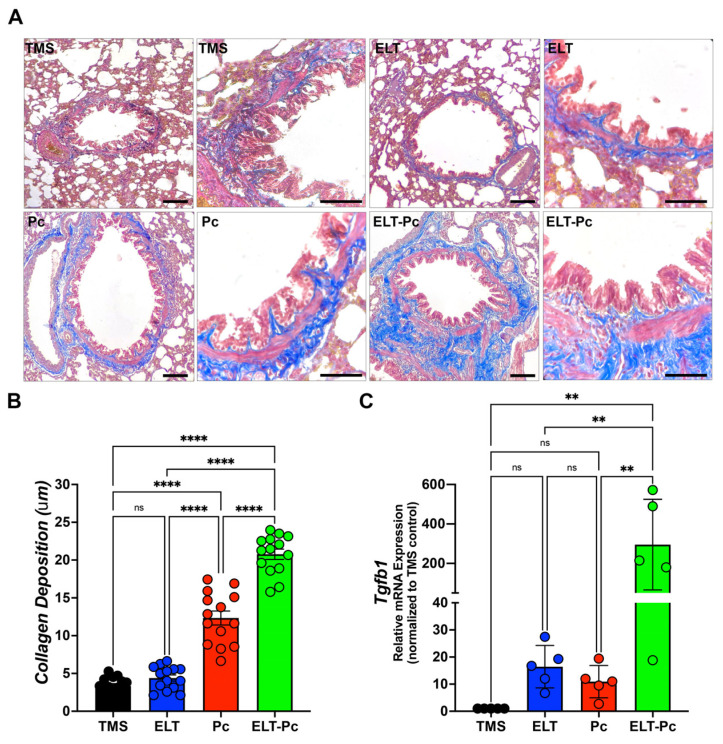
Evaluation of the collagen deposition around airways in the elastase-induced COPD animal model infected with *Pneumocystis*. (**A**) Masson trichome-stained lung sections from the experimental groups of rats were examined microscopically at 100× and 600× magnification to identify the presence of collagen deposition around airways. The bar indicates 100 μm in 100× magnification and 50 μm in 600× magnification. The quantification of collagen deposition was performed as described in the Section 4. (**B**) The quantification of collagen deposition is presented. Data are expressed as mean ± SD and analyzed by one-way ANOVA. Ns = no-significative; **** = *p* < 0.0001. (**C**) *Tgfβ1* mRNA levels were evaluated by qPCR. Data were determined using the 2^−DDCt^ method; actin levels were used as internal control and normalized to the saline control animal group. Data are expressed as mean ± SD and analyzed by ANOVA. Ns = non-significant; ** = *p* < 0.01.

**Table 1 ijms-25-03150-t001:** Primers used in this work.

Gene		Sequence (5′—3′)	Size (bp)
*Dhfr*	Forward	GTTGCACTTACAACTTCTTATGG	223
	Reverse	TAGATCCAGAGATTCATTTCGAG	
*Actin*	Forward	CTTGCAGCTCCTCCGTCGCC	228
	Reverse	CTTGCTCTGGGCCTCGTCGC	
*Il1b*	Forward	AGGCTTCCTTGTGCAAGTGT	71
	Reverse	TGTCGAGATGCTGCTGTGAG	
*Cox-2*	Forward	GATTGACAGCCCACCAACTT	187
	Reverse	CGGGATGAACTCTCTCCTCA	
*Tgfβ1*	Forward	AGGGCTACCATGCCAACTTC	111
	Reverse	CCACGTAGTAGACGATGGGC	
*PMuc5b (CRE)*	Forward	GGTGCCGGTATTTGTCTTTG	325
	Reverse	TGGGTGACTTCAGTATCATGG	

## Data Availability

The data presented in this study are available on request from the corresponding author. The data are not publicly available due to may be relevant to a future applied science project.

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
