# Peer review of "Role of the Fungus Pneumocystis in IL1β Pathway Activation and Airways Collagen Deposition in Elastase-Induced COPD Animals"

_ijms, 2024, doi:10.3390/ijms25063150_

Round 1

Reviewer 1 Report

Comments and Suggestions for Authors

The research manuscript “Role Of The Fungus Pneumocystis In IL1beta Pathway Activation And Airways Collagen Deposition In Elastase-induced COPD Animals” describes the involvement of the IL-1beta pathway in a model of COPD induced with elastase. Activation of the pathway after infection with the fungus Pneumocystis results in higher expression of MUC5B and deposition of collagen around the airways in rats. This is an interesting topic and it deserves further study. The manuscript is well written, the experiments well designed and clearly described. However, there are some concerns which need to be addressed:

Major concerns:

1: Figure 4D: How can you perform multiple comparisons and still have significance, with only five samples per group? The methods state that you performed adjustment for multiple comparisons. This cannot be true with only five measurements per group. Please explain.

Please adjust for multiple comparisons. This is applicable to all statistical tests used.

2: Line 168: “Animals instilled with elastase and infected with the fungus (ELT and Pc) exhibited an increment in mucus secretion in the airway epithelium compared to the control”. Based on your experiment, you cannot say anything about mucus secretion. You have measured mucus inside the goblet cells. Please rephrase. If you want to say anything about secreted mucus, you must fix your lung samples in Carnoy and quantify extracellular mucus.

3: Figure 5A: If it is correct that you viewed the sections in figure 2A in 100x magnification (likely if you used the 10x objective and had an additional 10x in the microscope optics), then you cannot have used 10 and 60x in figure 5A. The objectives may have been 10 and 60x, but the magnification was not. In the methods section, you write: The number of cysts was quantified using 20 fields per animal at a total magnification of 1000X. This cannot be true. At this magnification, you cannot quantify cysts. Please doublecheck the methods.

4: Please explain at first mention what the acronyms Pc and ELT-Pc stand for. It should be stated what they are acronyms of without looking at the figure.

5: In figure 3, you have two samples per group and in the file with the images of the whole blots, you have only included the blots in the figure. There are five points in each group in the quantification. Please include all blots used for quantification in the supplemental file.

Minor concerns:

1: A fungus cannot be called “who”. Please refer to it as “which” or “that”.

2: Reference 34 seems to be formatted wrong. Please revise.

Comments on the Quality of English Language

The language is good, with some minor  mistakes.

Author Response

Major concerns:

1: Figure 4D: How can you perform multiple comparisons and still have significance, with only five samples per group? The methods state that you performed adjustment for multiple comparisons. This cannot be true with only five measurements per group. Please explain.

Please adjust for multiple comparisons. This is applicable to all statistical tests used.

RESPONSE: Thank you for your comments. Specifically, the results presented in Figure 4D is associated with chromatin immunoprecipitation. Each sample is referred to the chromatin of each animal. That is the reason that is referred to as 5 samples per group. However, ChiP experiments were repeated at least three times (biological replicates), and the qPCRs to determine the enrichment were performed in triplicates (technical replicates). Finally, the means for each animal corresponded to each sample. Then, one-way ANOVA was performed. As post hoc test, multiple comparison analyses were performed. In this point, the software GraphPad Prism 10 recommends the test Tukey, which includes an adjustment of data using statistical hypothesis testing. Another adjustment such as Bonferroni was not recommended for our data. The other evaluations presented in the manuscript were performed in the same manner, adding biological and technical replicates to the analysis evaluating the means in each animal of each experimental group.

2: Line 168: “Animals instilled with elastase and infected with the fungus (ELT and Pc) exhibited an increment in mucus secretion in the airway epithelium compared to the control”. Based on your experiment, you cannot say anything about mucus secretion. You have measured mucus inside the goblet cells. Please rephrase. If you want to say anything about secreted mucus, you must fix your lung samples in Carnoy and quantify extracellular mucus.

RESPONSE: Thank you for the comment. The line indicated was changed in the text replacing the words “mucus secretion” with “presence of mucus”.

3: Figure 5A: If it is correct that you viewed the sections in figure 2A in 100x magnification (likely if you used the 10x objective and had an additional 10x in the microscope optics), then you cannot have used 10 and 60x in figure 5A. The objectives may have been 10 and 60x, but the magnification was not. In the methods section, you write: The number of cysts was quantified using 20 fields per animal at a total magnification of 1000X. This cannot be true. At this magnification, you cannot quantify cysts. Please doublecheck the methods.

RESPONSE: Thank you for the detection of this mistake. In the legend of Figure 5A, 10X and 60X were replaced by 100X and 600X according to the correct magnification used in the microscopy.

For the quantification of Pneumocystis cysts, the size of each cyst is around 10 mm. Therefore, is possible to detect cysts using 1000X magnification, because the size of the visual field diameter is 400 mm. Thus, it is possible to determine cysts in fields visualized using 1000X magnification. However, the detail of each cyst, for example, to identify nuclei if the cyst is empty, I understand that these details cannot be determined using this magnification. That is the reason for the determination of the fungus burden by qPCR to complement this quantification.

4: Please explain at first mention what the acronyms Pc and ELT-Pc stand for. It should be stated what they are acronyms of without looking at the figure.

RESPONSE: Thank you for the suggestion. In the text of the revised version of the manuscript, we have added the definition of each acronym associated with the experimental animal groups in the first paragraph of the results section.

5: In figure 3, you have two samples per group and in the file with the images of the whole blots, you have only included the blots in the figure. There are five points in each group in the quantification. Please include all blots used for quantification in the supplemental file.

RESPONSE: Thank you for the comment. We have added the other blots as a supplementary figure.

Minor concerns:

1: A fungus cannot be called “who”. Please refer to it as “which” or “that”.

RESPONSE: Thank you for the correction. We have corrected this point in the text.

2: Reference 34 seems to be formatted wrong. Please revise.

RESPONSE: Thank you for your comment. This reference was generated automatically using the software ZOTERO. We have corrected the text in an adequate format.

Reviewer 2 Report

Comments and Suggestions for Authors

In this paper Coronado and co-workers  investigated the contribution of Pneumocystis to the enhancement of inflammation, mucus secretion and fibrosis related to the induction of the IL1b and TGFb1 pathways within a COPD host. They conducted histological and molecular characterizations, assessing inflammatory, mucus and fibrosis-related markers in an rat elastase-induced COPD model.

The paper is well written and detailed in every part, I only have two minor comments: 

1. The limitations of the study are missing from the discussion. The authors should point out that this is an animal study mimicking the development of COPD/emphysema. The authors should aslo higlithed that  the disadvantage of the elastase model is that the function of elastase in COPD emphysema depends on several pathophysiological mechanisms which again brings up the number of clinical events.

2. The abstract should be more concise. Perhaps, the authors could think of a graphic abstract to make the message crisper.

Comments on the Quality of English Language

No comment. 

Author Response

The paper is well written and detailed in every part, I only have two minor comments: 

  1. The limitations of the study are missing from the discussion. The authors should point out that this is an animal study mimicking the development of COPD/emphysema. The authors should also highlighted that  the disadvantage of the elastase model is that the function of elastase in COPD emphysema depends on several pathophysiological mechanisms which again brings up the number of clinical events.

RESPONSE: Thank you for your comment. We have added a paragraph in the discussion section summarizing these aspects of the animal model and their limitations.

  1. The abstract should be more concise. Perhaps, the authors could think of a graphic abstract to make the message crisper.

RESPONSE: Thank you for your suggestion. We have elaborated a most concise abstract.

Reviewer 3 Report

Comments and Suggestions for Authors

Dear authors,

I read your manuscript concerning an in vivo rat model to investigate the involvement of the IL-1 pro-inflammatory pathway during Pneumocystis infection of the airways in COPD. In this paper, you report an update of your previous studies. The paper is clear, concise and reports a well-described investigation. Some points should be addressed.

1. The full name should precede abbreviations, check all the main text.

2.      MMP2 and MMP9 are involved in the degradation of collagen IV. Explain their role in the lungs and COPD.

3. What is endothelin? explain

4.      Line 236, remove « the human version of fungus », maybe the causative agent of human Pneumocystis infection

5.      Study limitations are missing. Add them to the discussion section.

6.      The discussion focuses on different aspects, but you should report the statistical significance. For example, in the Quantification of peribronchiolar and perivascular inflammation cuffs in the elastase-induced COPD model infected with Pneumocystis, there is no statistical significance between ELT, ELT, Pc and ELT-pc. The most exciting thing is collagen deposition and its clinical implications. The evaluation of mRNA is not always related to the production of the target protein report.

7.      High plagiarism detection was reported in the paper. You suggest you to edit the text.

8.      Liter, L please.

9.      What is the clinical implication of these results?

10.  To improve the introduction and clinical perspective, you should report current guidelines regarding pneumocystis pneumonia and the role of pneumocystis in the airways' mycobiota. 

11. Retinoids and retinoic acid pathways have been studied as promising molecules in the treatment and prevention of fibrosis and fungal diseases. Interestingly, retinoids have been shown to favour a positive outcome in PcP patients. I suggest reading and citing the following paper to improve discussion and perspective:

- Cosio T, Gaziano R, Zuccari G, Costanza G, Grelli S, Di Francesco P, Bianchi L, Campione E. Retinoids in Fungal Infections: From Bench to Bedside. Pharmaceuticals (Basel). 2021 Sep 24;14(10):962. doi: 10.3390/ph14100962. PMID: 34681186; PMCID: PMC8539705.

Comments on the Quality of English Language

Minor editing of English language required

Author Response

Dear authors,

I read your manuscript concerning an in vivo rat model to investigate the involvement of the IL-1 pro-inflammatory pathway during Pneumocystis infection of the airways in COPD. In this paper, you report an update of your previous studies. The paper is clear, concise and reports a well-described investigation. Some points should be addressed.

  1. The full name should precede abbreviations, check all the main text.

RESPONSE: Thank you for your suggestion. We have added the full names of some abbreviations. However, there are several standard abbreviations such as cytokines or other genes that we think it is not necessary to add their full names.

  1. MMP2 and MMP9 are involved in the degradation of collagen IV. Explain their role in the lungs and COPD.

REPONSE: Thank you for your comment. We have added a pair of lines indicating the role of these metalloproteases in the progression of COPD also adding a new reference.

  1. What is endothelin? Explain

RESPONSE: Thank you for your comment. This aspect was corrected in the introduction section.

  1. Line 236, remove « the human version of fungus », maybe the causative agent of human Pneumocystis infection

RESPONSE: Thank you for your suggestion. We have removed the text from the suggested phrase.

  1. Study limitations are missing. Add them to the discussion section.

RESPONSE: Thank you for your comment. We have added a paragraph in the discussion section indicating the limitations of studying COPD features in an elastase model.

  1. The discussion focuses on different aspects, but you should report the statistical significance. For example, in the Quantification of peribronchiolar and perivascular inflammation cuffs in the elastase-induced COPD model infected with Pneumocystis, there is no statistical significance between ELT, ELT, Pc and ELT-pc. The most exciting thing is collagen deposition and its clinical implications. The evaluation of mRNA is not always related to the production of the target protein report.

RESPONSE: Thank you for your suggestion. We have added a paragraph indicating the discussion of the non-significative differences in the number of cuffs between animal groups. An additional paragraph was added to complement our findings with clinical implications associated with collagen deposition and COPD.

  1. High plagiarism detection was reported in the paper. You suggest you to edit the text.

RESPONSE: Thank you for your comment. We have taken part in other papers of our authorship that could be repeated in other papers, especially the methodology section. However, we have analyzed and edited this manuscript to reduce possible plagiarism.

  1. Liter, L please.

RESPONSE: We do not understand this comment.

  1. What is the clinical implication of these results?

RESPONSE: Thank you for your comment. We have added an additional paragraph at the end of the discussion section commenting on the clinical impact of our findings.

  1. To improve the introduction and clinical perspective, you should report current guidelines regarding pneumocystis pneumonia and the role of pneumocystis in the airways' mycobiota. 

RESPONSE: Thank you for your comments. We have added these aspects in the introduction and discussion sections.

  1. Retinoids and retinoic acid pathways have been studied as promising molecules in the treatment and prevention of fibrosis and fungal diseases. Interestingly, retinoids have been shown to favour a positive outcome in PcP patients. I suggest reading and citing the following paper to improve discussion and perspective:

- Cosio T, Gaziano R, Zuccari G, Costanza G, Grelli S, Di Francesco P, Bianchi L, Campione E. Retinoids in Fungal Infections: From Bench to Bedside. Pharmaceuticals (Basel). 2021 Sep 24;14(10):962. doi: 10.3390/ph14100962. PMID: 34681186; PMCID: PMC8539705.

RESPONSE: thank you for your suggestion. We have added a little paragraph commented the role of RA pathway in the development of emphysema and the promising role of retinoids as anti-Pneumocystis drugs.